# A pathway towards high-quality development of the manufacturing industry: Does scientific and technological talent matter?

Dan LI[ORCID], Qiuyu YAO*

School of Business Administration, Liaoning Technical University, Huludao, Liaoning, China

* 1403959041@qq.com

**Data Availability Statement:** All relevant data are within the paper and its Supporting information files.

**Funding:** This study was supported by "Study on Countermeasures to Promote Talent Gathering and

## Abstract

Against the background of the accelerated evolution of the new round of scientific and technological revolution and industrial change, scientific and technical talents, as essential innovation resources, play an important role in promoting the high-quality development of the manufacturing industry. Based on the panel data of 30 provinces in China from 2012 to 2021, the article constructs a fixed-effects model and systematically researches the impact of scientific and technological talents on the high-quality development of the manufacturing industry. The results show that scientific and technical talents play a significant role in promoting the high-quality development of the manufacturing industry, and the upgrading of the consumption structure and the accumulation of productive service industries play a mediating role. Heterogeneity analysis found that the promotion effect of scientific and technical talents is more favorable in the eastern region, medium-technology level manufacturing, and labor-intensive manufacturing. Among the three sub-dimensions of scientific and technological talents, the scale of scientific and technical talents has the most significant impact on the development of the manufacturing industry. The analysis of the spatial spillover effect finds that scientific and technological talents will have a positive spillover effect on the development of the manufacturing industry in neighboring areas. The study provides a basis for relevant departments to formulate effective strategies and policies.

## 1. Introduction

The outline of the construction of a robust quality country proposes to speed up the implementation of the strategy of a vital quality country and speed up the iteration and quality upgrade of traditional manufacturing technology, promoting the quality of industrial products to the middle and high-end. This reflects the country's great importance to the construction of quality power and the country's confidence and determination to promote the high-quality development of manufacturing. China's manufacturing industry has been developing astonishingly since the reform and opening up. From a total output perspective, China has surpassed the United States to become the largest manufacturing nation, established itself as the world's most stable, complete industrial system, and played an irreplaceable role in the

Returning to Liaoning Based on College Students' Employment and Entrepreneurship in Liaoning" (2023lslwzzkt-019). The funders had no role in study design, data collection and analysis, decision to publish, or preparation of the manuscript.

**Competing interests:** The authors have declared that no competing interests exist.

international industrial division of labor [1]. The added value has consistently ranked first worldwide for 12 consecutive years. China's economy has entered the stage of high-quality development, and the manufacturing industry, the backbone of the industry, the foundation of the economy, and the pillar of people's livelihood can provide an essential guarantee for China's security and economic and social development. To achieve high-quality development of the manufacturing industry, which is a crucial element in building a modern socialist country and promoting high-quality economic growth, it is also an inevitable requirement to consolidate the competitive advantage of the manufacturing industry, enhance the stability and security of the industrial chain, and promote It is an unavoidable requirement for the manufacturing industry to advance to the middle and high end of the global value chain and is of great significance for building a strong manufacturing country and a strong quality country and promoting high-quality economic development. However, it can be seen from the World Bank statistics that the proportion of China's manufacturing value added to GDP has been fluctuating slightly near the peak after reaching the peak in 2006 and has been declining significantly since 2011, from 32.06% to 26.3% in 2020, with an average annual decline of 0.64 percentage points. This means that, compared to other developed countries, the decline rate of the proportion of the manufacturing industry in China is more prominent. In addition, China's manufacturing sector still has many flaws when compared to developed nations in terms of product variety and quality, including an absence of high-end, high-quality goods, severe homogeneous competition, a lack of market segmentation, a "neck" in crucial core technology, and a lack of international high-end brands. Then, an important issue that needs to be resolved is how to make up for these short boards to realize the intelligent, green, and high-end development of the manufacturing industry. In essence, talents, primarily scientific and technological talents, play an essential role in innovation and development and are the key to promoting the high-quality development of the manufacturing industry. Scientific and technical talents enable the green and intelligent transformation of the manufacturing industry and realize the high-quality development of the manufacturing industry by enhancing independent innovation ability and researching cutting-edge knowledge and technology.

At the 2021 Central Conference on Talent Work, General Secretary Xi Jinping pointed out that talent is the mainstay of national development and revitalization. It is necessary to focus on independent talent cultivation to enhance the country's competitive advantage in talent. The report of the 20th Party Congress also further emphasizes that cultivating a large number of high-quality talents with both virtue and talent is an excellent plan for the country's future and the nation's long-term development for a thousand years. Under the support and guidance of national strategy, with the deepening of education reform and the implementation of a series of education policies, the scale of China's manufacturing labor force has steadily expanded, the level of talent has continuously improved, the development mode has gradually shifted from "population dividend" to "talent dividend," and the effectiveness of talents has constantly improved. Human capital has become the leading force in promoting high-quality economic development, enhancing international competitiveness, and building an innovative country. However, driven by the new round of technological revolution and industrial change, China's manufacturing industry is gradually moving from the middle and low end to the high end, with a higher demand for high-level human capital, and the talent gap has become a key factor limiting the high-quality development of the manufacturing industry. The 2023 Global Digital Technology Development Study points out that although the total number of digital technology talents in China ranks first in the world, accounting for 17% of the global total, there are only 7,000 high-level digital technology talents, accounting for only 9% of the worldwide total. However, there are 21,000 in the United States, and China is only 35% of the United States, lagging far behind. The Planning Guide for the Development of Manufacturing Talents

points out that the shortage of manufacturing talents in critical areas in China will reach 30 million in 2025. At the same time, with the rise of new industries and new business models in high-end manufacturing, new talent gaps have been created, such as in the "China Artificial Intelligence Talent Training White Paper," which points out that the talent gap in China's artificial intelligence industry has reached 5 million people, which is a severe impediment to innovative development. Innovation is the first driving force of manufacturing development, and talent is a vital strategic resource for innovation. Promoting high-quality development in the manufacturing industry lies in the complete mobilization of the enthusiasm, creativity, and motivation of scientific and technological talent, and we must give full play to the supporting role of talent as the first resource. Scientific and technological talents' intellectual resources and innovative ideas are essential to the margin of innovation potential. Scientific and technical talents, as the main body of scientific and technological innovation and the engine driving economic and social development, are of great significance in improving productivity, developing the economy, pushing society forward, building an innovative country, helping China seize the opportunity in the new round of technological revolution, and promoting the realization of the goal of manufacturing power. Therefore, in the process of high-quality development of China's economy, it is of great theoretical significance and practical value to deeply study whether scientific and technological talents can promote the high-quality development of the manufacturing industry and how scientific and technical talents can influence the high-quality development of the manufacturing industry.

At present, scholars have conducted many useful explorations of the relationship between human capital and manufacturing development. Some scholars, such as Lucas (1988) [2], Romer (1990) [3], and Ciccone (2009) [4], from the perspective of "learning by doing", suggested that the improvement of labor efficiency and technology spillover caused by human capital in the process of production and material accumulation have positive effects on the upgrading of the manufacturing industry. Later, scholars explored the impact of human capital on manufacturing development from different dimensions. It has been shown that human capital accumulation [5–8] and human capital structure optimization [9–11] have a facilitating effect on manufacturing development. Chang et al. (2017) [12] explored the role of human capital in manufacturing plant growth using Taiwan as a research sample. Arjun et al. (2020) [13] empirically examines the effects of energy, human capital, finance, and technology on manufacturing value added in an endogenous growth framework. Shamsuzzoha et al. (2020) [14] examine the role of top managers' human capital and other exogenous determinants of the efficiency of manufacturing firms in Bangladesh by using heteroscedastic single-step stochastic frontier analysis. Dwikat et al. (2023) [15] studied the impact of competent human capital on manufacturing development. They found that the increase in the level of innovative human capital inhibits, to some extent, the positive effect of external knowledge sourcing on high-quality innovation in manufacturing. In addition, scholars have found that human capital can promote the development of the manufacturing industry through technological innovation [16–19], improving production efficiency [20, 21], improving consumption structure [22], and promoting the accumulation of productive service industries [23]. Under the impetus of the new industrial revolution, innovative human capital, scientific and technological talents, and other high-level human capital have gradually become hot spots for scholars' attention. Lin et al. (2021) [24] examined the relationship between innovative human capital and economic growth in China. Based on Lucasian endogenous growth theory, Xu et al. (2020) [25] use panel data models and spatial econometric methods to explore the relationship between innovative human capital and provincial economies. Wang et al. (2021) [26] studied the coupled and coordinated development of innovative human capital and green economic growth. Hu et al. (2021) [27] used spatial coupling analysis, cointegration, and the Granger

causality test model to analyze the relationship between scientific and technological talents and economic growth in the Pan-Yangtze River Delta and their spatial differences from a geographical perspective from 1998 to 2019. Zhang et al. (2015) [28] used mathematical modeling methods to analyze the intrinsic mechanism of technology talent clustering for economic growth. He et al. (2023) [29] found a significant positive correlation between the concentration of scientific and technological talents and the quality of economic growth in each city and that the level of regional economic development is the core factor influencing the flow of talent. In the era of the knowledge economy, high-level talents become an important force in promoting the innovation and transformation of the manufacturing industry, and their exploratory and breakthrough innovations promote the industry's high-quality development. Li et al. (2022) [30] pointed out that innovative human capital accumulation enhances the positive effect of GVC embedding on product upgrading and efficiency upgrading. Hou et al. (2023) [31] investigate the mechanism of the impact of external knowledge sourcing on high-quality innovation in manufacturing from the perspective of innovative human capital and find that the increase in the level of innovative human capital inhibits the positive effect of external knowledge sourcing on high-quality innovation in manufacturing to a certain extent. Gao et al. (2020) [32] explored the impact of scientific and technological talents, total factor productivity, and their interaction terms on the high-quality development of the equipment manufacturing industry based on human capital theory.

In summary, the existing literature has conducted in-depth research on human capital and the high-quality development of the manufacturing industry. However, there are still some things that could be improved. Firstly, previous researchers have focused on the discussion of the relationship between human capital level, human capital stock, human capital structure optimization, and manufacturing development, and only some scholars have paid attention to the role of scientific and technological talents in the development of high-quality manufacturing. The existing literature has yet to explore whether there is heterogeneity in the role of scientific and technical talents in manufacturing industries of different technology levels and various factor-intensive manufacturing industries. Secondly, the current studies measure the talent level by single indicators such as talent scale and education level, and scholars still need to explore the influence of the comprehensive development level of scientific and technological talents on the high-quality development of the manufacturing industry. Thirdly, the existing literature has mainly analyzed the role of technological innovation in developing the talent-influenced manufacturing industry, with less research on other influencing mechanisms. Lastly, although a small amount of literature has focused on the spatial spillover effect of talent on economic growth, it has yet to consider the spatial spillover effect of talent on the high-quality development of the manufacturing industry. Because of this, based on the panel data of 30 provinces in China from 2012 to 2021, this paper adopts the double fixed-effects model to explore whether scientific and technological talents can promote the high-quality development of the manufacturing industry and further analyzes in-depth the mediating effect of the upgrading of the consumption structure and the accumulation of the productive service industry, as well as the heterogeneity of the influence of scientific and technological talents and spillover effects, to provide ideas for the optimization of the structure of the distribution of the scientific and technical talents and the enhancement of the level for those areas where the high-quality development of the manufacturing industry and the scientific and technological talent level in the region have differences.

The marginal contributions of this paper are as follows: (1) From the perspective of scientific and technological talents, we analyze in depth whether and how scientific and technical talents can promote the high-quality development of the manufacturing industry from both theoretical and empirical aspects and explore whether there are differences in the role of

scientific and technological talents in the high-quality development of the manufacturing industry at different technical levels and different factor intensities. (2) Regarding index measurement, the index system of science and technology talent development is constructed from three dimensions: science and technology talent scale, science and technology talent environment, and science and technology talent effectiveness. The level of science and technology talent development in each region is measured comprehensively by the entropy value method to provide a more robust research basis for empirical analysis. (3) In terms of research content, it not only explores how scientific and technological talents influence the high-quality development of the manufacturing industry but also further explores the mechanism of the role of scientific and technological talents in influencing the high-quality development of the manufacturing industry and the spillover effect of scientific and technical talents from two aspects: consumption structure upgrading and productive service industry aggregation, to comprehensively measure the possible transmission channels of the influence of scientific and technical talents on the high-quality development of the manufacturing industry. (4) Considering the differences in manufacturing development level, economic development level, and policy measures in different regions, the heterogeneity of the influence of scientific and technological talents on the high-quality development of the manufacturing industry in the East, Middle, and West areas are explored to provide practical empirical support and decision-making reference for regional governments to promote the high-quality development of the manufacturing industry.

## 2. Theoretical analysis and research hypothesis

### 2.1. Promotional effect of scientific and technological talents on the high-quality development of the manufacturing industry

Schultz's human capital theory points out that among the many factors of production affecting economic growth, human capital is one of the most critical factors, and the knowledge, technology, and ability workers possess have a decisive role in economic development [33]. China has entered the stage of high-quality development, and the demand for high-quality human capital, such as scientific and technological talents, is more robust and urgent. As the core driving force of scientific and technological innovation, scientific and technical talents are essential in promoting the manufacturing industry's innovation and transformation and realizing the manufacturing industry's intelligent, green, and high-end development. High-quality enterprises are the basis of the high-quality development of the manufacturing industry, and high-level talents are one of the critical elements of innovation and transformation in high-quality enterprises [34]. Scientific and technological talents are more capable of absorbing and applying new knowledge and skills than ordinary employees. They can give full play to their external effects, transfer their rich knowledge reserves to skilled grassroots employees, promote the dissemination and application of new technologies, enhance the overall staff quality, drive the innovation of manufacturing enterprise management mode, promote the improvement and upgrading of the manufacturing production process, and finally improve the innovative production efficiency, resource allocation efficiency, and energy utilization efficiency of manufacturing enterprises, enhance the productivity and green development efficiency of the industry, and realize the green development of the manufacturing industry [35]. Using high-end technology and equipment, scientific and technological talents can improve manufacturing intelligence and help enterprises create new products, improve production processes, promote product quality upgrading and product structure optimization, reduce production costs, and enhance business efficiency. In addition, the flow of scientific and technological talents from inefficient sectors to high-efficiency sectors can promote the development of knowledge-

intensive industries, inter-industry division of labor, and the degree of refinement of manufacturing industries, realize the optimization and upgrading of industrial structure, further promote the high-quality development of manufacturing industries, and realize the construction of a strong manufacturing country. Based on the above analysis, the following hypotheses are proposed in this paper:

**Hypothesis 1.** Scientific and technological talents significantly contribute to the manufacturing industry's high-quality development.

## 2.2. The transmission mechanism of scientific and technological talents to the high-quality development of the manufacturing industry

**2.2.1. Consumption structure.** As a kind of high-level talent, technological talents have a higher income level than ordinary employees, and accordingly, their consumption level will also rise. Scientific and technical talents promote the high-quality development of the manufacturing industry by improving the consumption structure. According to Maslow's Hierarchy of Needs theory, when people's bottom physiological needs are satisfied, the proportion of spending on food and other necessities will gradually decrease, and people's consumption needs will shift from low-end products to high-end products. The diversification of consumer demand and the upgrading of consumption structure will force manufacturing enterprises to continuously improve the production process and structure [36], and promote the gradual optimization of the industrial layout, thus promoting the high-quality development of the manufacturing industry. In addition, with the improvement of the consumption level and the upgrading of the consumption structure, enterprises continue to improve the production process and develop new products to meet the latest consumer demand, thus increasing the supply of new products and achieving the optimal industrial configuration of the whole market [37]. Driven by the new demand, the manufacturing enterprises' production and operation, technology application, and product supply capabilities are continuously improved, thus forming a scale effect, prompting a more effective allocation of resources and a significant reduction in production costs, and promoting the high-quality development of the manufacturing industry. Further, the following hypothesis is formulated:

**Hypothesis 2.** Scientific and technological talents indirectly contribute to the high-quality development of the manufacturing industry by improving the regional consumption structure.

**2.2.2. Agglomeration of productive service industries.** Combined with the theory of industrial clusters, the accumulation of productive service industries can produce strong agglomeration, scale, spillover, and competition effects [38, 39], thus impacting the manufacturing industry's high-quality development. The improvement of the level of scientific and technological talents means that the attractiveness of regional talents is enhanced, and the production service industry, as a knowledge-intensive industry, will also experience the phenomenon of agglomeration. Manufacturing enterprises can obtain external benefits through accumulation, accelerate the overflow and creation of knowledge and technology, and finally promote the high-quality development of the manufacturing industry by improving the technological innovation and efficiency of the manufacturing industry, reducing the production cost of enterprises, and improving the added value of enterprises' products [40]. In addition, the geographical concentration of the production service industry is accompanied by the relative concentration of capital, technology, information, equipment, and other factors, which

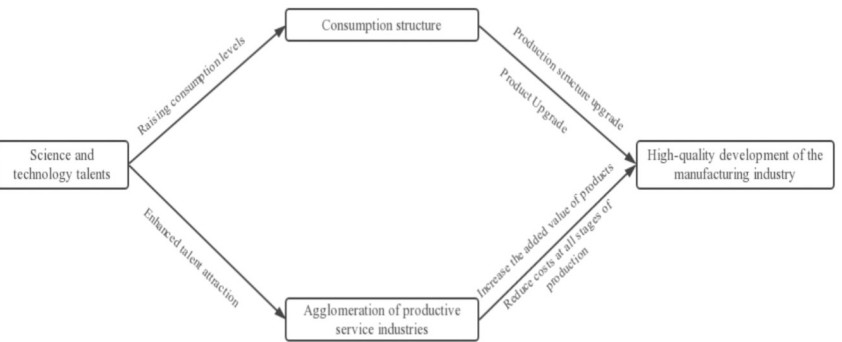

**Fig 1. A theoretical model of intermediation effects.**

effectively reduces the cost of each link in manufacturing production [41] and thus promotes the competitiveness of enterprises, which is ultimately beneficial to the development of the manufacturing industry. Based on this, hypothesis 3 is suggested:

**Hypothesis 3.** Scientific and technological talents indirectly promote the high-quality development of the manufacturing industry by improving the level of concentration of the regional productive service industry.

The transmission mechanism of scientific and technological talents to the high-quality development of the manufacturing industry is shown in Fig 1.

## 2.3. The Spatial spillover effect of scientific and technological talents

With the construction of a sizeable domestic circulation pattern and the deepening of the reform of the household registration system, the resistance to inter-regional factor flow reduces, and innovation factors such as scientific and technological talents flow more smoothly between regions, which in turn has an impact on the development of manufacturing industries in other areas. We dissect the spatial spillover effect of scientific and technological talents on the high-quality development of the manufacturing industry in two aspects: First, the positive spillover effect. For developed regions, their good living and working environments, sufficient production and innovation resources, and perfect policies and systems have a "siphon effect" on scientific and technological talents, which makes the regional stock of scientific and technical talents increase continuously. As a kind of high-level human capital, scientific and technological talents have more advanced knowledge and technology stock, prompting the region to enhance its innovation capacity, continuously promote technological progress, and drive the high-quality development of the regional manufacturing industry by improving production processes, increasing innovative production efficiency, and optimizing management models. As the number of regional scientific and technological talents increases, the competitiveness of those talents will be enhanced accordingly, and some scientific and technical talents will choose to go to regions with relatively weaker competition for employment, thus generating spatial spillover effects. In addition, the knowledge and skills of scientific and technological talents have certain externalities, and economically backward regions can promote the technological progress of the region by learning from the innovation achievements of developed areas and absorbing spillover effects, thus promoting the high-quality development of the manufacturing industry in the region. Second, the negative spillover effect. The location decision of scientific and technological talents is not only influenced by various factors such as

welfare compensation, cost of living, technological innovation capacity, and economic development level of the region but also by the results of cross-sectional comparisons with neighboring regions. Regions with similar geographical locations and economic levels compete more fiercely for innovation resources such as scientific and technological talents, and areas with developed economic levels, abundant resources, and strong scientific and technical innovation capabilities are more competitive. The lagging regions need help attracting high-quality production factors and face the challenge of the region's resources being attracted by relatively developed areas, harming the high-quality development of the region's manufacturing industry. Therefore, what kind of spillover effect scientific and technological talents will have on the high-quality development of the manufacturing industry needs to be further explored. Based on the above analysis, the following hypotheses are proposed in this paper:

**Hypothesis 4.** There is a positive spillover effect of scientific and technological talents on the high-quality development of the manufacturing industry in neighboring areas.

**Hypothesis 5.** There is a negative spillover effect of scientific and technological talents on the high-quality development of the manufacturing industry in neighboring regions.

## 3. Study design

### 3.1. Model construction

Benchmark regression model. In order to verify whether scientific and technological talents can promote the high-quality development of the manufacturing industry, we construct the following benchmark regression model:

$$Mhf_{it} = \alpha_0 + \alpha_1 Rdt_{it} + \alpha_c Z_{it} + \mu_i + \delta_t + \varepsilon_{it} \tag{1}$$

In Eq (1), i represents the province, and t represents the year. $Mhf_{it}$ and $Rdt_{it}$ correspond to the level of high-quality development of the manufacturing industry and the level of development of scientific and technological talents, respectively. $Z_{it}$ is the ensemble of control variables, $\varepsilon_{it}$ represents the random disturbance term, and $\mu_i$ and $\delta_t$ denote regional and time-fixed effects.

Spatial econometric model. Considering the possible delay in the role of scientific and technological talents in the development of the manufacturing industry and the existence of spatial dependence on the high-quality development of the manufacturing industry itself, we construct the following spatial Durbin model to verify the spillover effect of the scientific and technological talents:

$$Mhf_{it} = \alpha_0 + \alpha_1 W_{it} Mhf_{it} + \alpha_2 Rdt_{it} + \alpha_3 W_{it} Rdt_{it} + \alpha_4 Z_{it} + \alpha_5 W_{it} Z_{it} + \mu_i + \delta_t + \varepsilon_{it} \tag{2}$$

In Eq (2), $W_{it}$ is the spatial weight matrix, and the geographic distance matrix is selected considering that the neighboring provinces have more frequent cooperation and exchange; $\alpha_0$ is the constant term, $\alpha_1$ represents the spatial autoregressive coefficient of high-quality development of the manufacturing industry, $\alpha_2$ is the coefficient of the development level of scientific and technological talents, $\alpha_3$ and $\alpha_5$ represent the coefficients of the influence of scientific and technical skills talents from other regions on the observed values of the area, and $\alpha_4$ is the coefficient of control variables.

Intermediary effects model. This study refers to Jiang's study (2022) [42] for testing the transmission mechanism of scientific and technological skills affecting the high-quality growth of the manufacturing industry to avoid the issues of endogeneity and inadequate identification of some channels in the three-step test of mediating effects. Firstly, analyzing the impact of

scientific and technological talents on the high-quality development of manufacturing based on Eq (3). Secondly, according to the analysis of theoretical mechanisms, the impact of the upgrading of consumption structures and the accumulation of productive service industries on the high-quality development of the manufacturing industry is significant. Thirdly, testing whether the effect of scientific and technological talents on the mediating variable is substantial according to Eq (4).

$$Mhf_{it} = \alpha_0 + \alpha_1 Rdt_{it} + \alpha_2 Z_{it} + \mu_i + \delta_t + \varepsilon_{it} \tag{3}$$

$$C_{it} = \beta_0 + \beta_1 Rdt_{it} + \beta_2 Z_{it} + \mu_i + \delta_t + \varepsilon_{it} \tag{4}$$

In Eqs (3) and (4), $C_{it}$ represents the ensemble of mediating variables, including consumption structure upgrading, and productive service industry agglomeration, $\beta_1$ indicates the influence of scientific and technical talents on mediating variables, and the meanings of the remaining variables are consistent with Eq (1).

### 3.2. Variable description

**3.2.1. Explained variables.** The explanatory variable in this paper is the high quality of the manufacturing industry's development level (*Mhf*). There are two main approaches to measuring quality manufacturing development: One is using a single indicator to measure the variable, such as manufacturing total factor productivity [43–45] or industry value added rate, manufacturing output per capita, industrial value added, etc. [46]; The other is using the rating index system, a comprehensive measure of high-quality manufacturing development [47–53]. The high-quality development of the manufacturing industry is a multi-dimensional and comprehensive concept, including the improvement of science and technology innovation, structural improvement, economic efficiency, factor allocation optimization, product quality enhancement, continuous promotion of the integration of the two, and continuous improvement of the green level. In order to make the measurement results of manufacturing high-quality development level more objective and comprehensive, based on the connotation and characteristics of manufacturing high-quality development, the evaluation index system of manufacturing high-quality development, containing seven first-level indicators and twenty-three second-level indicators, as shown in Table 1. Further, we use the entropy value method to measure each index's data and use the total score obtained to measure the level of high-quality development in the manufacturing industry.

**3.2.2. Explanatory variables.** The explanatory variable in this paper is the level of development of scientific and technological talents (*Rdt*). Most existing studies measure scientific and technological talents by a single indicator, such as Ma (2019) [54], which measures scientific and technical talents through employees' education and disciplines. Throughout the foreign science and technology innovation evaluation system, there are relatively few independent evaluation index systems for science and technology talents, and most of the science and technology talent evaluation indexes are integrated into the index system for science and technology innovation development.

Combining theoretical analysis and scientific and technological innovation development practice at home and abroad and referring to existing research results [55], this paper constructs a multi-level and multi-dimensional scientific and technical talent development index system to comprehensively reflect each province's current scientific and technological talent development situation. In order to grasp the overall situation of science and technology talent development, judge the future development trend of science and technology talent, and examine the contribution and efficiency of science and technology talent, our study constructs an

**Table 1. Manufacturing quality development level evaluation index system.**

| Tier 1 Indicators | Secondary Indicators | Description of specific indicators |
|---|---|---|
| Innovation-driven | Innovation Inputs | Personnel input |
| | | Financial investment |
| | Innovation Output | Number of patents |
| | | Value added of new products |
| Structural Optimization | Industry Structure | Degree of industrialization |
| | Enterprise Structure | The proportion of high-tech enterprises |
| | Product Structure | The balance of new high-tech products |
| | Export Structure | Export value of high-tech industry |
| Speed Effect | Growth Rate | The growth rate of industrial value added |
| | Profitability | Main business profit to primary business income |
| | Production cost | The ratio of main business cost to primary business income |
| | Assets and liabilities | Assets and liabilities ratio |
| Factor Effect | Labor efficiency | Revenue from main business operations per capita |
| | Capital efficiency | Return on Assets |
| Quality Brand | Quality | Product quality qualification rate. |
| | Brand | The percentage of the number of top 500 manufacturing enterprises |
| Integrated Development | Integration of Two Chemicals | Cell phone penetration rate |
| | | Internet penetration rate |
| | | E-commerce application level |
| Green Development | Environmental Governance | Investment in pollution control |
| | Pollution Emission | Wastewater emissions |
| | | Exhaust gas emissions |
| | | Solid Waste Emissions |

evaluation index system for the development of scientific and technological talents containing 25 positive indicators from three dimensions of science and technology talent investment, science and technology talent environment, and science and technology talent effectiveness, and further measures the level of science and technology talent development in each region by the entropy value method. The details are shown in Table 2.

**3.2.3. Mediating variables.** (1) The consumption structure (*Con*). Drawing on Wang's study (2022) [56], using the proportion of the sum of three types of consumption expenditures, transportation and communication, education and entertainment, and health care, in total consumption expenditures, to measure the consumption structure. (2) Productive service industry clustering (*Agg*). In productive service industry clustering, we use the location entropy index to measure the agglomeration level of the productive service industry in each region based on the number of people employed in the productive service industry. Referring to Kena's study (2017) [57], employment in finance, information transmission, computer services and software, scientific research, technical services, leasing, business services, and transportation are used as benchmarks for measurement, as shown in the following formula, where i represents provinces, j represents productive services, and e represents employment.

$$Agg_i = \left( \frac{e_{i,j}}{\sum_{j=1}^{m} e_{i,j}} \right) / \left( \frac{\sum_{i=1}^{n} e_{i,j}}{\sum_{j=1}^{m} \sum_{i=1}^{n} e_{i,j}} \right) \tag{5}$$

**Table 2. Science and technology talent development index system.**

| Tier 1 Indicators | Secondary Indicators | Description of specific indicators |
|---|---|---|
| Technology Talent Input | The Scale of Science and Technology Talents | Number of R&D personnel per 10,000 people |
| | | Number of R&D personnel in high-tech enterprises |
| | | Number of people employed in science and technology services |
| | | Number of professional and technical personnel |
| | | Number of Scientists and Engineers |
| | Science and Technology Talent Structure | The proportion of master's degree holders |
| | | Number of middle and senior title holders |
| Science and Technology Talent Environment | The attractiveness of Scientific and Technological Talents | Per capita financial income |
| | | Disposable income per capita |
| | | Per capita consumption expenditure |
| | | Average salary level of workers |
| | | Science and Technology Expenditure Ratio |
| | | Per capita Park green space |
| | | Number of enterprises with R&D institutions |
| | Science and Technology Talent Cultivation | The proportion of spending on education |
| | | Number of full-time teachers in general higher education schools |
| | Science and Technology Talent Reserve | Number of students in available higher education institutions |
| Science and technology talent capacity | Basic Research Capability | Number of scientific and technical papers published |
| | | Number of granted patents for inventions |
| | Technology Innovation Capability | Technology Market Turnover |
| | Industry-driven Capability | Number of new product projects |
| | | Main Business Income of High Technology Industry |

**3.2.4. Control variables.** In order to measure the effect of scientific and technological talent more accurately on the manufacturing industry's high-quality development and control for other factors that may affect the high-quality development of the manufacturing industry, we select control variables regarding the literature in this area. The degree of government intervention (*Gov*), expressed as the share of general government budget expenditure in regional GDP; The level of urbanization (*Ubr*), measured as the proportion of the urban population in the resident population of each province; The level of infrastructure (*Lnf*), expressed as the number of road miles built per capita; Fixed asset investment (*Inv*), described as the share of fixed asset investment in regional GDP in each province; Level of external openness (*Fdi*), expressed as the logarithm of actual utilization of foreign investment in each section; Level of financial development (*Fin*), measured as the sum of deposits and loans in each province as a share of GDP; Level of marketization (*Mar*), measured by drawing on the marketization index estimated by Yu et al. [58]; Industrial structure (*Is*), calculated by the ratio of the tertiary sector's output value to the secondary sector's output value. The main variables are defined in Table 3.

**Table 3. Variable definition.**

| Type | Symbol | Meaning | Definition |
|---|---|---|---|
| Explained variables | Mhf | The level of high-quality development in the manufacturing industry | Evaluation index system for high-quality development of the manufacturing industry |
| Explanatory variables | Rdt | The story of the development of scientific and technological talents | The evaluation system for the development of scientific and technical talents |
| Mediating variables | Con | Consumption Structure | (per capita transportation and communication expenditure + per capita education, culture and entertainment expenditure + per capita health care expenditure)/per capita consumption expenditure |
| | Agg | The concentration of the productive service industry | Calculation of industrial agglomeration using location entropy index |
| Control variables | Gov | Level of government intervention | Budget expenditure/GDP |
| | Ubr | Level of urbanization | Urban population/year-end resident population |
| | Lnf | Level of Infrastructure | Road mileage per square kilometer |
| | Inv | Fixed Asset Investment | Investment in fixed assets/GDP |
| | Fdi | Degree of opening to the outside world | ln (The actual amount of foreign investment utilized) |
| | Fin | Level of financial development | (Deposits + Loans)/GDP |
| | Mar | Marketization level | Marketization index |
| | Is | Industrial Structure | The output value of tertiary industry/output value of secondary industry |

## 3.3. Data sources and descriptive statistics of variables

Our study uses the panel data of 30 provinces in China (excluding Tibet and Hong Kong, Macao, and Taiwan) from 2012–2021 as the sample. We obtain the original data from the China Statistical Yearbook, the China Industrial Statistical Yearbook, the China Science and Technology Statistical Yearbook, and the China Labor Statistical Yearbook in previous years. Some missing data were obtained from the provincial statistical bulletins and statistical yearbooks, while the remaining missing values were filled by interpolation. In addition, to eliminate the dimensional relationship between variables, avoid numerical problems, and make the data more comparable, the data in non-ratio values is processed by taking logarithms. The results of the descriptive statistics for the variables are shown in the Table 4.

**Table 4. Descriptive statistics.**

| Variable Type | Variable Symbol | Sample size | Mean value | Standard deviation | Min value | Maximum value |
|---|---|---|---|---|---|---|
| Explained variables | Mhf | 300 | 0.198 | 0.113 | 0.076 | 0.755 |
| Explanatory variables | Rdt | 300 | 0.147 | 0.122 | 0.014 | 0.676 |
| Mediating variables | Con | 300 | 0.333 | 0.039 | 0.241 | 0.433 |
| | Agg | 300 | 0.850 | 0.072 | 0.693 | 1.073 |
| Control variables | Gov | 300 | 0.251 | 0.103 | 0.107 | 0.643 |
| | Ubr | 300 | 0.602 | 0.118 | 0.363 | 0.896 |
| | Lnf | 300 | 0.006 | 0.009 | 0.001 | 0.052 |
| | Inv | 300 | 0.803 | 0.263 | 0.205 | 1.480 |
| | Fdi | 300 | 5.448 | 1.772 | -1.575 | 7.722 |
| | Mar | 300 | 8.138 | 1.882 | 3.359 | 12.390 |
| | Fin | 300 | 3.331 | 1.150 | 1.568 | 8.131 |
| | Is | 300 | 1.283 | 0.711 | 0.549 | 5.297 |

# 4. Analysis of the empirical results

## 4.1. Baseline regression results

Table 5 reports the impact of scientific and technological talent on the high-quality development of the manufacturing industry. Column (1) shows the estimation results without considering any control variables or fixed effects, while column (2) shows the regression results of fixed individual and time effects, and the estimated coefficients are significantly positive, indicating that scientific and technological talents can significantly promote the high-quality development of the manufacturing industry. Column (3) adds control variables to column (2), and the results show that the regression coefficient of scientific and technological talent is positive and significant at the 1% level. The above results indicate that scientific and technological talents can significantly contribute to the high-quality development of the manufacturing industry. Thus, hypothesis 1 of this paper has been verified. This result is consistent with Gao's findings. By integrating and reconstructing knowledge and technology, technological talents transfer their professional and R&D knowledge to grassroots employees, thus improving overall workforce quality. In addition, their strong desire for knowledge and keen observation of technological talents enables them to quickly absorb advanced knowledge and technology and apply them to production operations [59], improve the technological innovation ability of enterprises, and then promote high-quality development of the manufacturing industry by enhancing production capacity [60], improving production processes, promoting product upgrading, and optimizing management modes.

**Table 5. Baseline regression results.**

| Variables | (1) | (2) | (3) |
|---|---|---|---|
| | *Mhf* | *Mhf* | *Mhf* |
| *Rdt* | 0.814*** | 0.820*** | 1.013*** |
| | (5.009) | (3.620) | (7.917) |
| *Gov* | | | 0.193** |
| | | | (2.558) |
| *Ubr* | | | 0.585*** |
| | | | (3.861) |
| *Lnf* | | | 3.829 |
| | | | (1.128) |
| *Inv* | | | -0.023*** |
| | | | (-4.210) |
| *Fdi* | | | -0.003 |
| | | | (-1.172) |
| *Mar* | | | -0.006** |
| | | | (-2.205) |
| *Fin* | | | -0.007 |
| | | | (-0.941) |
| *Is* | | | -0.047*** |
| | | | (-4.201) |
| _cons | 0.079*** | -0.122 | -0.413*** |
| | (4.751) | (-1.261) | (-3.064) |
| Fixed effects | No | Yes | Yes |
| N | 300 | 300 | 300 |
| $R^2$ | 0.770 | 0.972 | 0.983 |

Note:

*, **, *** are significant at 10%, 5% and 1% levels, respectively. Same after table.

This study further complements and extends existing research findings. We investigate the principle that scientific and technological talent promotes manufacturing innovation and transformation through knowledge spillover and technology spillover. Existing studies have analyzed the relationship between human capital and the development of the manufacturing industry through theory or empirical evidence, and they have argued that an increase in human capital stock, structural optimization, and level enhancement contribute to technological innovation, which improves productivity and reduces costs. In the context of the new round of the industrial revolution, our study further explores the mechanism of its impact on manufacturing development from the perspective of scientific and technological talents.

The regression results of the control variables show that: (1) The government's financial support and the improvement of the urbanization level can significantly promote the high-quality development of the manufacturing industry. (2) The increase in fixed asset investment, the elevation of marketization level, and the expansion of the tertiary sector will have a suppressive effect on the high-quality development of the manufacturing industry. (3) The impact of the infrastructure level and the degree of opening to the outside world on the high-quality development of the manufacturing industry needs to be further strengthened.

## 4.2. Endogenous discussion

Scientific and technological talents can promote the high-quality development of the manufacturing industry. At the same time, the improvement of the high-quality development level of the manufacturing industry will also enhance the competitiveness of regional scientific and technological talents, prompting the continuous expansion of the scale of scientific and technical talents and the continuous improvement of the effectiveness of scientific and technological talents, thus driving the elevation of the development level of regional scientific and technological talents. Considering that the potential two-way causality between scientific and technical talents and the high-quality development of the manufacturing industry may cause endogeneity problems and affect the regression, for this purpose, two-stage least squares estimation is performed by introducing lagged first and lagged second orders of technological talent. In addition, to avoid the endogeneity problem of lagging manufacturing high-quality development, a systematic GMM model is constructed for estimation by introducing the lagged first order of manufacturing high-quality development.

The F-values of the weak instrumental test in the two-stage least squares were 5637.37 and 2194.18, which were greater than 10, rejecting the hypothesis that the instrumental variables were weakly instrumental and indicating that the selected instrumental variables were reasonably valid. The estimation results of columns (1) to (4) in Table 6 show that the first-stage regression results indicate that the instrumental variables are significantly and positively related to the high-quality development of the manufacturing industry, and the second-stage regression results indicate that scientific and technological talents can distinctly promote the high-quality development of the manufacturing industry, which again verifies the regression results of the previous paper. The p-value of the tested AR (2) in the system GMM estimation is 0.77, and the p-value of the Hansen test is 1.00, both of which are greater than 0.1, which satisfies the condition of using the system GMM and indicates that the system GMM estimation results are valid and feasible. The estimated results in column (5) of Table 6 show that the lagged first-order coefficient of high-quality manufacturing development is significantly positive at the 1% level, indicating that the previous period's development level influences the current period's manufacturing development level. The estimated coefficient of scientific and technological talents is still significantly positive, which once again confirms the robustness of the baseline regression results. After considering the endogeneity issue, the scientific and

**Table 6. Endogeneity test results.**

| Explanatory variables | 2SLS | | | | (5) System GMM |
|---|---|---|---|---|---|
| | (1) First stage of return | (2) Second stage of return | (3) First stage of return | (4) Second stage of return | |
| Rdt | | 0.996*** (28.960) | | 1.003*** (29.340) | 0.462*** (1.725) |
| L.Rdt | 1.086*** (75.080) | | | | |
| LL. Rdt | | | 1.164*** (46.840) | | |
| L.Mhf | | | | | 0.668*** (2.622) |
| Control variables | Yes | Yes | Yes | Yes | Yes |
| Fixed effects | Yes | Yes | Yes | Yes | Yes |
| _cons | -0.011* (-1.690) | 0.190*** (6.770) | -0.022** (-2.080) | 0.232*** (7.770) | -0.670** (-2.350) |
| $R^2$ | 0.996 | 0.908 | 0.992 | 0.914 | |
| N | 270 | 270 | 240 | 240 | 270 |

technical talents still considerably contribute to the manufacturing industry's high-quality development, and hypothesis 1 still holds.

## 4.3. Robustness tests

To further test the reliability of the above findings, the robustness of the regression results was tested by replacing the explanatory variables, changing the sample period, removing some samples, and using quantile regression.

(1) Replace the core explanatory variables. We conduct the regression analysis again by replacing the original index of the development level of scientific and technological talents with the number of R&D personnel per 10,000 people. As shown in column (2) of Table 7, there is still a significant positive relationship between scientific and technological talent and high-quality development in the manufacturing industry. This indicates that the results are still robust after replacing the core explanatory variables.

(2) Replacement sample period. In order to avoid subjective bias from sample selection, we adjust the sample years in this paper. This study excludes 2012 and 2021 and selects the data from 2013 to 2020 for re-estimation. The regression results in column (3) of Table 7 show that

**Table 7. Robustness test results.**

| Variables | (1) | (2) | (3) | (4) | (5) | | |
|---|---|---|---|---|---|---|---|
| | | | | | 25% | 50% | 75% |
| Rdt | 1.013*** (7.917) | | 1.086*** (7.601) | 1.022*** (17.248) | 0.967*** (11.463) | 1.026*** (19.794) | 1.040*** (24.047) |
| Rdt1 | | 0.080*** (2.940) | | | | | |
| _cons | -0.413*** (-3.064) | 0.619 (1.570) | -0.505*** (-4.567) | 0.180 (1.422) | 0.116*** (4.290) | 0.140*** (5.906) | 0.161*** (3.498) |
| Control variables | Yes | Yes | Yes | Yes | Yes | Yes | Yes |
| Fixed effects | Yes | Yes | Yes | Yes | Yes | Yes | Yes |
| N | 300 | 300 | 240 | 260 | 300 | 300 | 300 |
| $R^2$ | 0.983 | 0.953 | 0.987 | 0.989 | 0.606 | 0.670 | 0.718 |

the estimated coefficients of the scientific and technological talent variables are still significantly positive. This indicates that other factors do not affect the regression results, suggesting robust conclusions.

(3) Delete the four municipalities directly under the central government. Compared with other regions, municipalities directly under the central government have a higher level of development of scientific and technological talents, which will amplify the effect of scientific and technical talents to promote the high-quality development of the manufacturing industry. For this reason, our study removes the four municipalities directly under the central government and conducts the regression analysis again. According to the regression results in column (4) in Table 7, scientific and technological talents can still significantly promote the high-quality development of the manufacturing industry, indicating that the benchmark regression results are robust and reliable.

(4) Quantile regression. We use the quantile regressions to examine further whether there are differences in the impact of scientific and technological talents on the high-quality development of the manufacturing industry at different development levels. The regression results from column (5) of Table 7 show that scientific and technological talents can significantly promote the high-quality development of the manufacturing industry in all three quartiles. The estimated coefficient of scientific and technical talents gradually increases as the quartiles increase, indicating that the higher the level of scientific and technological talents, the stronger its effect on promoting the high-quality development of the manufacturing industry, which again verifies the robustness of the regression results.

## 5. Further discussion

### 5.1. Analysis of the mechanism of action

The studies above show that scientific and technological talents can promote the high-quality development of the manufacturing industry. Then, what is the driving mechanism of scientific and technological talents for the high-quality development of the manufacturing industry? The previous paper analyzed how scientific and technical talents can indirectly promote the high-quality development of the manufacturing industry by fostering technological innovation, upgrading consumption structures, and agglomerating the productive service industry. In the following, we will use the mediating effect model to test this empirically and investigate how scientific and technological talents can influence the high-quality development of the manufacturing industry through the above three channels.

**5.1.1. Consumption structure.** From column (1) of Table 8, we find that scientific and technological talents can significantly promote the high-quality development of the manufacturing industry; Column (3) reports the effect of scientific and technological talent on

**Table 8. Mechanism of action analysis.**

| Variables | (1)<br>Mhf | (2)<br>Con | (3)<br>Agg |
|:---:|:---:|:---:|:---:|
| Rdt | 0.990***<br>(30.598) | 0.072**<br>(2.307) | 0.080**<br>(2.116) |
| _cons | 0.143***<br>(4.620) | 0.300***<br>(10.041) | 0.888***<br>(24.717) |
| Control variables | Yes | Yes | Yes |
| Fixed effects | Yes | Yes | Yes |
| N | 300 | 300 | 300 |
| $R^2$ | 0.904 | 0.241 | 0.676 |

consumption structure upgrading, and the regression results indicate that scientific and technological talent can significantly contribute to consumption structure upgrading. In addition, the previous theoretical analysis section specifies that the consumption structure upgrade significantly contributes to the manufacturing industry's high-quality development. In summary, consumption structure upgrades mediate the effect of scientific and technological talents on high-quality manufacturing development. Meanwhile, according to the results of the Bootstrap test, the 95% confidence interval is (0.002, 0.033), which does not contain a zero value, further confirming the existence of the mediating effect of consumption structure upgrading. Thus, hypothesis 2 of this paper is verified. Technological talents have higher income levels than other human capital and correspondingly higher consumption levels. People's consumption demand gradually shifts from low-end to high-end products, and the consumption structure upgrades continuously. In order to meet the diversified needs of consumers and adapt to the gradually improving consumption structure, enterprises must constantly improve production processes and optimize the industrial layout to improve the quality and supply capacity of their products. Then the resource allocation and production efficiency of manufacturing enterprises are improved, and production costs are significantly reduced to promote the high-quality development of the manufacturing industry.

**5.1.2. Productive service industry agglomeration.** The regression results of column (4) in Table 8 show a significant promotion effect of scientific and technological talents on the mediating variable productive service industry agglomeration. In addition, based on the theoretical analysis in the previous section, the improvement of the agglomeration level of the productive service industry has a prominent role in promoting the high-quality development of the manufacturing industry. In summary, effective service industry agglomeration plays a mediating effect in the role of scientific and technological talents in the high-quality development of the manufacturing industry. In addition, according to the results of the Bootstrap test, the 95% confidence interval does not contain a zero value, which again verifies the existence of an intermediary effect of accumulation in the productive service industry. Thus, hypothesis 3 of this paper is confirmed. The improvement of the level of scientific and technological talents will attract more high-level talents, promote the optimization of regional human capital structures, and then drive the concentration of the productive service industry, which can provide specialized and personalized services for manufacturing production, reduce enterprise information search and transaction costs, improve the international competitive advantage of the manufacturing industry through the economy of scale effect, agglomeration effect, and spillover effect, and promote the high-quality development of the manufacturing industry [39, 57].

Our study further extends the research of Anyanwu, Yang, et al. [5, 23]. They explained the role of human capital in manufacturing development from the perspective of technological innovation. Our empirical results show that scientific and technical talents promote the high-quality development of the manufacturing industry through the mediating effects of consumption structure upgrading and productive service industry agglomeration. The study is conducive to a comprehensive understanding of the role of scientific and technological talents in the transformation and development of the manufacturing industry, which is of great significance for improving the competitiveness of China's manufacturing industry.

## 5.2. Heterogeneity analysis

**5.2.1. Regional heterogeneity.** The differences in welfare policies, mechanisms and institutions, and economic levels of each region may affect the promotion effect of scientific and technological talents, resulting in differences in the impact of scientific and technical talents on the high-quality development of the manufacturing industry in each region. Therefore, the

**Table 9. Descriptive statistical analysis of variables in different regions.**

| Variables | Region | Sample size | Mean value | Standard deviation | Median |
|---|---|---|---|---|---|
| Mhf | Eastern Region | 110 | 0.281 | 0.141 | 0.230 |
| | Central Region | 80 | 0.170 | 0.053 | 0.160 |
| | Western Region | 110 | 0.137 | 0.043 | 0.120 |
| Rdt | Eastern Region | 110 | 0.243 | 0.149 | 0.210 |
| | Central Region | 80 | 0.113 | 0.043 | 0.100 |
| | Western Region | 110 | 0.075 | 0.041 | 0.060 |

30 provinces were divided into eastern, central, and western areas according to the National Bureau of Statistics division criteria to explore the differences in the effects of scientific and technological talents on the high-quality development of the manufacturing industry in different regions. Before conducting the categorical regression test, this study performs descriptive statistical analysis on the differences in the manufacturing industry's high-quality development level and the comprehensive development level of scientific and technological talents in different regions. As shown in Table 9, the spatial structure of the manufacturing quality development level and the science and technology talent level both show the characteristics of a vital East and a weak West. The eastern region is ahead of the central and western areas, which lays the foundation for the regional heterogeneity test below.

Table 10 shows the regression analysis of regional heterogeneity, and the results show that the estimated coefficients of scientific and technological talents in eastern, central, and western regions pass the significance test at the 1% level, indicating that scientific and technical talents in different areas can significantly promote the high-quality development of the manufacturing industry. Regarding the magnitude of the estimated coefficients, the promotion effect of scientific and technological talents shows a general trend of "east > west > central," with the promotion effect of scientific and technical talents in the eastern region being stronger. This may be due to the following reasons: (1) The east area has a developed economic level, a high degree of openness to the outside world, a high degree of manufacturing intelligence, a high degree of industrial clustering, and abundant educational resources. In the digital era, the substitution effect of production intelligence on basic labor is enhanced, and the demand for high-quality human capital is higher. With the improvement of the level of human capital, the manufacturing enterprises' technological innovation ability also improves continuously, and the efficiency of labor production and resource allocation is gradually improved, which promotes the transformation of the manufacturing industry into something green, intelligent, and high-end. (2) Although China has introduced a series of policies to support the development

**Table 10. Regression results of regional heterogeneity.**

| Variables | (1) Eastern Region | (2) Central Region | (3) Western Region |
|---|---|---|---|
| Rdt | 1.078*** | 0.394*** | 0.685*** |
| | (13.002) | (2.731) | (5.588) |
| _cons | -0.377*** | -0.174 | 0.127 |
| | (-2.653) | (-1.130) | (1.242) |
| Control variables | Yes | Yes | Yes |
| Fixed effects | Yes | Yes | Yes |
| N | 110 | 80 | 110 |
| $R^2$ | 0.983 | 0.971 | 0.954 |

of scientific and technological talents in recent years, the scale of these talents continues to expand, and their effectiveness of talents continues to be enhanced. However, the central region's investment in education and scientific research is low, forming a large gap with the eastern part. Especially in the Central Plains Economic Zone, although the investment in talent is significant, the overall level is low, the brain drain is severe, the institutional mechanisms are more fettered, and the efficiency of resource allocation could be better, hindering the talent's advantage. In addition, the level of intelligence of the manufacturing process in the central region is low, the dependence on basic labor is relatively high, and the degree of influence of the scientific and technological talents on the development of the regional manufacturing industry is limited. (3) Under the influence of the Western development strategy and innovation-driven development strategy, the Western region has continuously implemented talent introduction and cultivation programs, and the national and local governments have also formulated a series of policies and measures to encourage outstanding talents to go to remote and western regions for employment. The Western region has entered an era of rapid development with the help of national policies. The level of economic development and informationization has continued to rise. Educational resources have become increasingly abundant, and the level of economic development and information technology continues to increase. With the support of a series of policies, many high-end talents will choose to develop in the Western region, driving the level of scientific and technological talent in the area of the West. However, there is still a significant gap compared with the eastern region, and the promotion of manufacturing development is relatively weak.

**5.2.2. Heterogeneity in the sub-dimension of scientific and technological talent.**
Table 11 reports the regression results of the three sub-dimensions of scientific and technical talent affecting the high-quality development of the manufacturing industry. The regression coefficients of the technological talent scale, technological talent environment, and technological talent effectiveness are all significantly positive at the 1% level, indicating that the technical talent scale, technological talent environment, and technological talent effectiveness all significantly contribute to the high-quality development of the manufacturing industry. Further analysis reveals that the scale of scientific and technical talents has the most apparent promoting effect on the high-quality development of the manufacturing industry, followed by talent environment and talent effectiveness. The reasons for this difference may be: (1) Under the guidance of national strategic policies, China has increased the importance of primary research talent cultivation, and since the 19th National Congress, the scale and growth rate of scientific

**Table 11. Heterogeneity regression results by dimension.**

| Variables | (1) Technological talent scale | (2) Technological talent environment | (3) Technological talent effectiveness |
|---|---|---|---|
| $Rdt\_1$ | 3.451*** (5.139) | | |
| $Rdt\_2$ | | 2.089*** (10.285) | |
| $Rdt\_3$ | | | 1.569*** (3.047) |
| _cons | -0.761*** (-4.383) | 0.093 (0.524) | 0.305 (1.179) |
| Control variables | Yes | Yes | Yes |
| Fixed effects | Yes | Yes | Yes |
| N | 300 | 300 | 300 |
| $R^2$ | 0.978 | 0.981 | 0.964 |

and technological talents have increased significantly, the total number of R&D researchers has ranked first in the world for many years, the hierarchical structure has become increasingly reasonable, and the reserve team of scientific and technological talents has been expanding and improving in quality, providing strong talents for the development of the manufacturing industry. (2) The government has formulated a series of policies and measures to attract more high-level talent and build a high-quality and high-level talent team. However, there are still many things that need to improve in the talent introduction and management mechanism, evaluation and incentive mechanism, service guarantee and training system, etc., the global competitiveness of technological talent needs to be stronger, and thus the empowering effect on manufacturing development has declined. (3) Although the scale of output effectiveness of scientific and technical talents has increased significantly, the output intensity still needs to be higher, and there is a large gap between the output effectiveness in academic and economic aspects compared with that of developed countries. Thus, the promotion effect on the development of the manufacturing industry is relatively limited.

**5.2.3. Heterogeneity of manufacturing industries of different technology types.**
According to the criteria of the Organization for Economic Cooperation and Development for classifying manufacturing technology, we divide each manufacturing sub-sector into low-tech, medium-tech, and high-tech manufacturing industries and explore the heterogeneous characteristics of the impact of the level of scientific and technological talents on the high-quality development of manufacturing industries of different technology types. Table 12 shows that scientific and technical talents can promote the high-quality development of high-technology and low- and medium-technology manufacturing industries. Still, the degree of their influence on the development of the manufacturing industry at different technological levels is different. The most apparent effect of scientific and technical talents on the high-quality development of medium-technology manufacturing industries is followed by high-technology manufacturing industries, and the weakest effect is promoting the high-quality development of low-technology manufacturing industries. It may be because the medium and high technology manufacturing industry develops faster, has higher knowledge and technology intensity, and has higher requirements for production factors such as technology and talent. Scientific and technological talents master more advanced knowledge and technology and have higher creativity than ordinary employees, which can better meet the needs of transforming medium and high technology manufacturing industry development. Therefore, the higher the level of scientific and technical talents, the more knowledge and technology they master, the stronger their innovation ability, and thus the more significant the promotion of high-quality development in the manufacturing industry.

**Table 12. Regression results of heterogeneity of manufacturing industries with different technology types.**

| Variables | (1) High-technology | (2) Medium-technology | (3) Low-technology |
|---|---|---|---|
| $Rdt$ | 1.171*** (7.9245) | 1.186*** (8.7970) | 1.088*** (8.4430) |
| _cons | -0.910*** (-4.2206) | -0.944*** (-4.6242) | -0.800*** (-3.9159) |
| Control variables | Yes | Yes | Yes |
| Fixed effects | Yes | Yes | Yes |
| N | 300 | 300 | 300 |
| $R^2$ | 0.981 | 0.983 | 0.983 |

**Table 13. Regression results of heterogeneity of different factor-intensive manufacturing industries.**

| Variables | (1)<br>Labor-intensive | (2)<br>Capital-intensive | (3)<br>Technology-intensive |
|---|---|---|---|
| *Rdt* | 1.172***<br>(8.0631) | 1.169***<br>(7.9446) | 1.099***<br>(7.8031) |
| _cons | -0.863***<br>(-3.8431) | -1.004***<br>(-4.7094) | -0.771***<br>(-3.8417) |
| Control variables | Yes | Yes | Yes |
| Fixed effects | Yes | Yes | Yes |
| N | 300 | 300 | 300 |
| $R^2$ | 0.983 | 0.983 | 0.983 |

**5.2.4. Heterogeneity of different factor-intensive manufacturing industries.** In this paper, referring to the classification criteria of Care and Wang et al. (2023) [61], the manufacturing industry is divided into labor-intensive, capital-intensive, and technology-intensive segments. Then we examine the differences in the degree of influence of the development level of scientific and technological talents on the high-quality development of different factor-intensive manufacturing industries. As can be seen from Table 13, scientific and technical talents have apparent positive effects on the high-quality development of various factor-intensive manufacturing industries, among which scientific and technological talents have the most substantial promotion effect on the high-quality development of labor-intensive manufacturing industries, followed by capital-intensive ones, and the most miniature influence effect on the high-quality development of technology-intensive manufacturing industries. This may be because the labor-intensive manufacturing industry is more dependent on labor, and the rich knowledge reserve and advanced technology mastered by the scientific and technological talents themselves can significantly improve the innovative production efficiency of the labor-intensive manufacturing industry and promote the level of intelligent development of labor-intensive manufacturing industries to achieve their high-quality development.

Our study extends the findings of Gao et al. [32]. They argued that there are differences in the role of scientific and technological talents in promoting the high-quality development of the manufacturing industry in different regions. However, they only explored the impact of scientific and technological talent based on regional heterogeneity analysis. We analyze the heterogeneity of the effects of scientific and technological talent on the development of the manufacturing industry from different regions' perspectives, dimensions of core explanatory variables, different technological levels of the industry, and different factor intensities. Our study helps to fully understand the role of scientific and technical talents in developing the manufacturing industry. It provides a basis for the relevant departments to formulate strategies in a targeted manner.

## 5.3. Analysis of spatial spillover effects

**5.3.1. Spatial correlation analysis.** Under the promotion of a regionally coordinated development strategy, the inter-regional connection is getting closer and closer, and the level of scientific and technological talent and the level of development of the manufacturing industry in one area may influence the development of the manufacturing industry in other regions. Therefore, the spatial measurement method is further adopted to explore this phenomenon. We must examine the spatial autocorrelation of variables before conducting a spatial regression analysis. First, under the geographic matrix, we calculate the global Moran's I indices of

**Table 14. Global Moran's I of independent and dependent variables.**

| Year | Mhf | | Rdt | |
|---|---|---|---|---|
| | Moran's I | Z-value | Moran's I | Z-value |
| 2012 | 0.067*** | 2.865 | 0.060*** | 2.689 |
| 2013 | 0.068*** | 2.910 | 0.060*** | 2.676 |
| 2014 | 0.071*** | 2.968 | 0.061*** | 2.738 |
| 2015 | 0.074*** | 3.086 | 0.064*** | 2.797 |
| 2016 | 0.071*** | 2.998 | 0.058*** | 2.633 |
| 2017 | 0.056*** | 2.608 | 0.046** | 2.293 |
| 2018 | 0.048*** | 2.424 | 0.040** | 2.149 |
| 2019 | 0.054*** | 2.609 | 0.041** | 2.155 |
| 2020 | 0.055*** | 2.645 | 0.042** | 2.197 |
| 2021 | 0.043** | 2.300 | 0.047*** | 2.322 |

high-quality development of science and technology talents and the manufacturing industry from 2012–2021. The results are shown in Table 14, the global Moran's I values of high-quality development of the manufacturing industry, as well as science and technology talent, were significantly positive, and the Z values were all greater than 1.96; Secondly, the local Moran's I indices of high-quality development of science and technology talents and manufacturing industry in 2012 and 2021 were calculated, and it can be seen from Figs 2–5 that the Moran's indices of high-quality development of science and technology talents and manufacturing industry are primarily distributed in the first and third quadrants, indicating that the high-quality development of manufacturing industry and the level of science and technology talents in each region have high spatial correlation and spatial clustering characteristics.

**5.3.2. Selecting a spatial econometric model.** The results of the above study suggest that spatial regression analysis can be performed. Then, which model to choose needs to be further tested. Firstly, the results of the LM test show that the P-values of the LM-lag and LM-error tests are less than 0.1, which tentatively suggests that the spatial Durbin model is the most suitable; Secondly, according to the results of the Hausman test, the P-value is significantly 0,

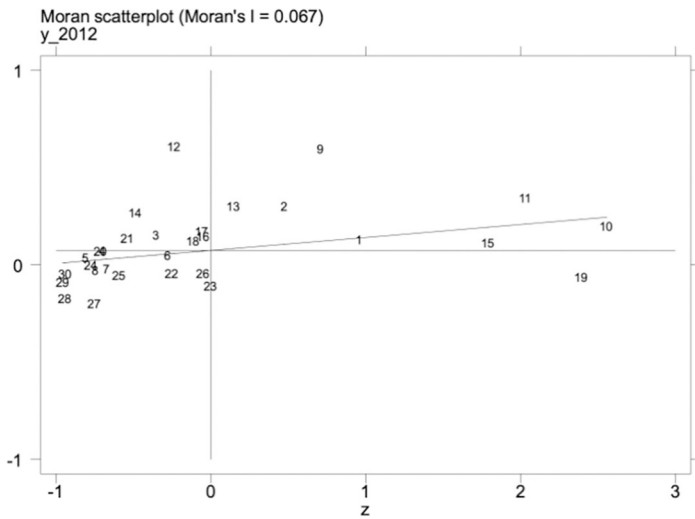

**Fig 2. Scatterplot of Moran index of science and technology talent development in 2012.**

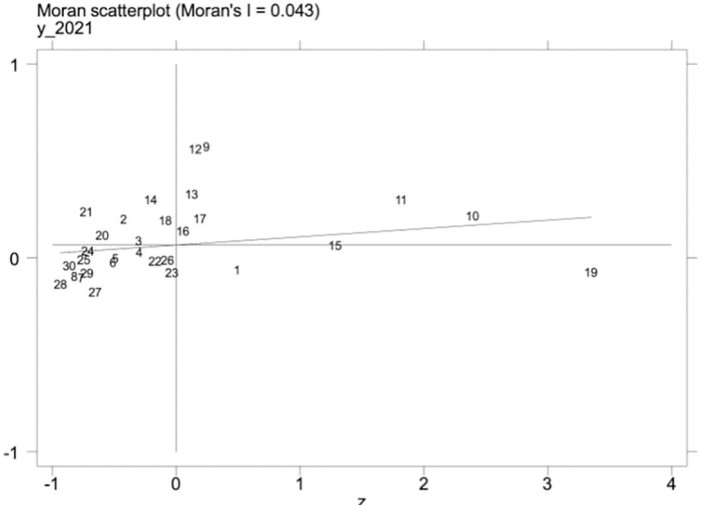

**Fig 3. Scatterplot of Moran index of science and technology talent development in 2021.**

which indicates that this study should select the two-way fixed-effect model; Finally, according to the results of the LR test, we find that the spatial Durbin model is better than the spatial lag model and The spatial error model, therefore, the two-way fixed spatial Durbin model is finally selected to study the spatial spillover effect of scientific and technological talents on the high-quality development of the manufacturing industry.

**5.3.3. Spatial spillover effect analysis.** In order to ensure the robustness of the empirical results, our study explores the spatial spillover effects of scientific and technological talents on the high-quality development of manufacturing under three spatial measurement models (SDM, SAR, and SEM) under the adjacency matrix and the geographical distance matrix. The regression results in Table 15 show that the spatial autoregressive coefficient of high-quality manufacturing development under the geographic distance matrix is 0.177 and significant at

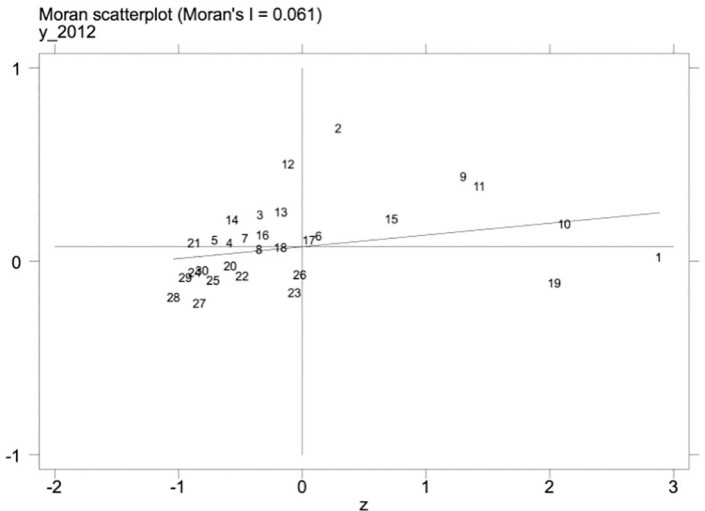

**Fig 4. Scatterplot of Moran index of high-quality manufacturing development in 2012.**

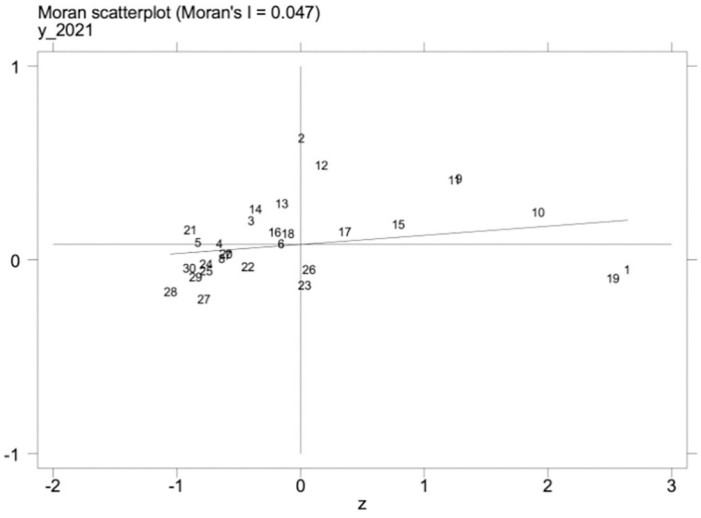

**Fig 5. Scatterplot of Moran index of high-quality manufacturing development in 2021.**

the 10% level, indicating that high-quality manufacturing development in one region positively affects manufacturing development in neighboring areas to a certain extent. The coefficient of the spatial lag term of scientific and technological talents is significantly positive under the geographic distance matrix, indicating a spatial spillover effect of the development of scientific and technical talents in the region on the high-quality development of the manufacturing industry in neighboring areas. However, the simple point regression results do not explain the spillover effect of scientific and technological talents on the high-quality development of the manufacturing industry, so the impact of scientific and technical talents on the high-quality development of the manufacturing industry is further analyzed from two perspectives: direct

**Table 15. Regression results of the spatial model.**

| Variables | Adjacency matrix | | | Geographical distance matrix | | |
|---|---|---|---|---|---|---|
| | SAR | SEM | SDM | SAR | SEM | SDM |
| *Rdt* | 1.004*** (24.307) | 0.952*** (31.697) | 1.016*** (22.569) | 0.990*** (24.526) | 0.955*** (30.266) | 1.039*** (24.959) |
| *W*Rdt* | | | 0.117 (1.088) | | | 0.257* (1.700) |
| Control variables | Yes | Yes | Yes | Yes | Yes | Yes |
| Direct effect | 1.007*** (23.633) | | 1.018*** (21.993) | 1.002*** (23.920) | | 1.055*** (23.741) |
| Indirect effect | 0.096** (2.412) | | 0.147** (2.006) | 0.294*** (2.724) | | 0.537*** (3.175) |
| Total effect | 1.103*** (17.176) | | 1.166*** (12.210) | 1.295*** (10.563) | | 1.592*** (8.490) |
| Rho | 0.088*** (2.645) | | 0.024 (0.308) | 0.228*** (3.504) | | 0.177* (1.841) |
| lambda | | 0.345*** (5.132) | | | 0.553*** (6.916) | |
| Sigma_e | 0.000*** (12.220) | 0.000*** (11.396) | 0.000*** (12.250) | 0.000*** (12.222) | 0.000*** (11.211) | 0.000*** (12.211) |
| N | 300 | 300 | 300 | 300 | 300 | 300 |

effect and indirect effect. The results show that scientific and technological talents' direct and indirect effects on manufacturing high-quality development are significantly positive under both the adjacency matrix and the geographical distance matrix, confirming the positive spillover effect of scientific and technical talents on manufacturing high-quality development in neighboring regions. With the smoother flow of talent and other factors, the competition among regions on production factors such as talent becomes more intense. Still, due to the spillover effect of knowledge and technology, the transfer of technology, senior human capital, and management experience is promoted through the flow of scientific and technological talents, knowledge and technology exchange, and capital circulation, which is conducive to manufacturing technology innovation, improving production efficiency and resource allocation efficiency, and optimizing structural layout, thus promoting neighboring regions' high-quality development of the manufacturing industry. From the above, it can be seen that hypothesis 4 holds.

The existing literature does not analyze the spatial spillover effect of human capital on manufacturing development. Based on the spatial econometric model, we explored the spillover effect of scientific and technological talents on the high-quality development of the manufacturing industry. The development of scientific and technical talents can promote the transformation and upgrading of manufacturing industries in neighboring regions, thus realizing the improvement of national manufacturing competitiveness. Our study provides a theoretical basis for regional governments to formulate policies and for the central government to coordinate and plan the development of the manufacturing industry in each region, which will help to reduce the differences in the level of manufacturing industry development in each region.

## 6. Conclusions and discussions

### 6.1. Conclusions

Based on the perspective that scientific and technological talents play an essential role in the innovation and transformation of the manufacturing industry, this paper starts from the viewpoint of technological innovation, consumption structure upgrading, and productive service industry agglomeration, takes the panel data of 30 provinces in China from 2012 to 2021 as the research samples, explores the effect of scientific and technological talents on the high-quality development of manufacturing industry based on the fixed-effect model, and further empirically analyzes the driving mechanism and spillover effect of scientific and technical talents on the high-quality development of manufacturing industry through the mediating effect model and spatial Durbin model. The following conclusions are drawn from the study: (1) Scientific and technological talents obviously promote the high-quality development of the manufacturing industry and have become the key to promoting innovation and transformation of the manufacturing industry, and the promotion effect of scientific and technological talents still holds through robustness tests such as replacing explanatory variables, changing the sample period, deleting some samples, and quantile regression. (2) The results of the mediating effect indicate that scientific and technological talents will indirectly promote the high-quality development of the manufacturing industry by promoting consumption structure upgrading and productive service industry clustering. (3) The heterogeneity analysis shows that the level of high-quality development of the manufacturing industry and the overall level of scientific and technological talents in the Eastern region are higher than those in the central and Western areas, and the scientific and technical talent dividend enjoyed by the Eastern region is relatively more apparent. The promotion effect of the scale of scientific and technological talents on the high-quality development of the manufacturing industry is the strongest among the

three dimensions of the scientific and technical talent development index system. The heterogeneity analysis of different technology types and factor-intensive manufacturing industries shows that the promotion effect of scientific and technological talents on medium-technology and labor-intensive manufacturing industries is the strongest. (4) The spatial spillover effect analysis shows that the high-quality development of scientific and technical talents and the manufacturing industry in this region promotes the development of the manufacturing industry in neighboring regions through spatial spillover.

## 6.2. Theoretical implications

This study provides important theoretical implications for the existing literature in three ways. First, although a large body of literature has investigated the impact of human capital stock, human capital accumulation, human capital mobility, and human capital structure optimization on economic growth and industrial development, there are few studies on the relationship between scientific and technological talent and manufacturing development. Most existing studies use the number of R&D personnel to represent the scientific and technological talent variable, which cannot comprehensively measure the level of scientific and technological talent. Therefore, this study measures the level of scientific and technological talents by constructing the evaluation index system of scientific and technical talents' development level, and on this basis, explores the promotion effect of scientific and technological talents on the high-quality development of the manufacturing industry.

Second, although the government and enterprise managers have made many efforts to improve the overall talent level to break through the intelligent, green, and high-end development dilemma faced by the manufacturing industry, it is still unclear how scientific and technological talent affects the transformation and upgrading of the manufacturing industry by improving the consumption structure of the population and promoting the accumulation of productive service industries. Our study hypothesizes and empirically confirms the two channels through which scientific and technological talents influence the high-quality development of the manufacturing industry: upgrading the consumption structure and accumulating productive service industries.

Third, existing studies have explored the differences in the impact of human capital on manufacturing development from different regional perspectives to provide a basis for formulating relevant policies in each region. Still, there needs to be more research on the heterogeneity of the impact effect of scientific and technological talent at different technical levels and different factor intensities. Therefore, this study analyzes heterogeneity at different technology levels and at different factor intensities based on the analysis of regional heterogeneity in previous literature. Our study finds that the role of scientific and technological talents in developing medium-technology and labor-intensive manufacturing industries is more prominent, providing new perspectives and ideas for subsequent research.

## 6.3. Policy and managerial implications

Based on the study results, we put forward several suggestions to promote the development of the manufacturing industry, which provide references for the government to formulate relevant policies and are of great practical significance.

First, pay attention to the role of scientific and technological talents in promoting the high-quality development of the manufacturing industry and improving the level of scientific and technological talents through multiple channels. Increase the government's investment in scientific and technical talents, build a few essential discipline training bases, strengthen the infrastructure construction of higher education institutions and the level of teachers, and improve

the scientific and technological talent training system that meets the requirements of national strategies and industrial needs. Deepen the reform of the evaluation mechanism for scientific and technical talents and establish a perfect evaluation system for scientific and technological talents. Reform the science and technology education model, focus on the new model of compound professional training, and encourage universities to expand education content to cultivate science and technology talents that meet the needs of national strategy implementation and industrial development. In addition, it is necessary to pay more attention to cultivating young scientific and technological talents, develop and implement special research projects for elemental talents, and support the development of promising young talents. Increase the opening of talents to the outside world, encourage and guide domestic scientific and technological talents to go to top international research institutions for cooperation and exchange, and at the same time, strengthen the work of introducing talents to attract more scientific and technological talents to return to China. Through the above channels, we will turn the constant talent advantage into surging and inexhaustible development power, lay the talent foundation for the construction of manufacturing power, and promote the high-quality development of China's manufacturing industry.

Second, pay attention to the role of consumption structure upgrading and productive service industry agglomeration in promoting the high-quality development of the manufacturing industry. Promote the upgrading of consumption structure, establish a sound regulatory system and consumer rights protection mechanism, build a perfect credit system, meet the economic and social development and diversified needs of consumers, stimulate people's consumption potential, build an olive-shaped distribution structure to narrow the gap between the rich and the poor, enhance consumers' purchasing power, thus promoting the upgrading of consumption levels and forcing manufacturing enterprises to transform and upgrade; Improve the agglomeration level of the productive service industry, strengthen the policy guidance and financial support for the development of the influential service industry, establish agglomeration demonstration areas, build public service platforms, and adjust the internal structure of the productive service industry according to market demand to realize the positive interaction between the productive service industry and the high-quality development of the manufacturing industry.

Third, accelerate the flow of scientific and technological talents, knowledge, and technology between regions, radiate and drive the development of manufacturing industries in neighboring areas, and give full play to the spatial spillover effect of scientific and technological talents and high-quality development of manufacturing industries. Accelerate the construction of transportation infrastructure, provide transportation convenience for the flow of scientific and technical talents between regions, accelerate the construction of new infrastructure, establish an online talent exchange platform, and promote the sharing and exchange of talents, knowledge, technology, and other resources between regions; Deepen the reform of the household registration system, break regional affiliation, departmental boundaries, and identity restrictions, and reduce the flow constraints of scientific and technological talents; Promoting the construction of an integrated public service system for the flow of talents, the strengthening of the informationization of public services for talent mobility, and the improvement of the level of convenience of public services for the mobility of scientific and technological talents will help the mobility of scientific and technical talents between regions and unleash the overflow effect of scientific and technological talents.

Fourth, because of the differences between different regions in terms of economic development level, working and living environment, the scale of scientific and technological talents, demand for manufacturing development and policy measures, etc., the policy planning related to the development of scientific and technical talents and high-quality development of the

manufacturing industry is reasonably formulated. For the eastern region, it should continue to give full play to the advantageous conditions, cultivate scientific and technological talents prospectively, and strengthen the flow of scientific and technical talents while promoting the high-quality development of the manufacturing industry in this region to drive the high-quality development of the manufacturing industry in the neighboring areas. For the central and western regions, we should adopt the "two-pronged" strategy of cultivating scientific and technological talents in the area and introducing talents, strengthening the investment in education and scientific research, improving the construction of "double first-class" universities, the structure of teachers and infrastructure, and establishing a sound and diversified teaching mode. We should adopt multiple preferential policies, strengthen the investment in public services such as medical and health care, continuously optimize the system and innovation environment, create a good research atmosphere for the growth and role of talents, and give scientific and technological talents more autonomy in research to enhance the competitive advantage of regional talents, attract more high-level talents to the region, and promote the development of regional manufacturing innovation.

## 6.4. Limitations and future research directions

There are several limitations to this study. First, the driving role of scientific and technological talents in the development of the manufacturing industry may also be affected by factors other than the control variables selected in this paper. More influencing factors can be introduced into the study in the future by continuously improving the data acquisition and measurement methods. Second, due to the limitation of data availability, this paper explores the impact of scientific and technological talents on the high-quality development of the manufacturing industry only at the provincial level. The research scale is relatively large, and the author will improve it in future research by summarizing the relevant scientific and technological talent data into city data and then conducting research on cities.

## Supporting information

**S1 Data.**
(XLSX)

## Acknowledgments

The author gratefully acknowledges the editors and referees for their positive and constructive comments in the review process.

## Author Contributions

**Conceptualization:** Dan LI, Qiuyu YAO.

**Data curation:** Qiuyu YAO.

**Formal analysis:** Qiuyu YAO.

**Funding acquisition:** Dan LI.

**Investigation:** Dan LI, Qiuyu YAO.

**Methodology:** Dan LI.

**Resources:** Qiuyu YAO.

**Software:** Dan LI.

**Supervision:** Dan LI.

**Validation:** Dan LI.

**Visualization:** Dan LI.

**Writing – original draft:** Qiuyu YAO.

**Writing – review & editing:** Dan LI, Qiuyu YAO.

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
