## [Decision Letter · Decision Letter 0]

13 Jul 2023

PONE-D-23-20007Study on the impact of scientific and technological talents on high-quality development of manufacturing industry: An empirical study based on provincial panel data.PLOS ONE

Dear Dr. YAO,

Thank you for submitting your manuscript to PLOS ONE. After careful consideration, we feel that it has merit but does not fully meet PLOS ONE’s publication criteria as it currently stands. Therefore, we invite you to submit a revised version of the manuscript that addresses the points raised during the review process.

We look forward to receiving your revised manuscript.

Kind regards,

William Mbanyele, PhD

Academic Editor

PLOS ONE

Journal Requirements:

   "This research was funded by “Research on the Driving Mechanism and Policy Options of Digital Transformation of Liaoning Manufacturing Industry” (No. L22BJY032."

  "This research was funded by “Research on the Driving Mechanism and Policy Options of Digital Transformation of Liaoning Manufacturing Industry” (No. L22BJY032) "

   "This research was funded by “Research on the Driving Mechanism and Policy Options of Digital Transformation of Liaoning Manufacturing Industry” (No. L22BJY032) "

6. Please amend the manuscript submission data (via Edit Submission) to include author Dan LI.

7. Please remove your figures from within your manuscript file, leaving only the individual TIFF/EPS image files, uploaded separately. These will be automatically included in the reviewers’ PDF.

8. We note you have included a table to which you do not refer in the text of your manuscript. Please ensure that you refer to Table 2, 4, 14 and 15 in your text; if accepted, production will need this reference to link the reader to the Table.

Reviewers' comments:

Reviewer's Responses to Questions

**Comments to the Author**

1. Is the manuscript technically sound, and do the data support the conclusions?

Reviewer #1: Yes

2. Has the statistical analysis been performed appropriately and rigorously? 

Reviewer #1: Yes

3. Have the authors made all data underlying the findings in their manuscript fully available?

Reviewer #1: Yes

4. Is the manuscript presented in an intelligible fashion and written in standard English?

Reviewer #1: Yes

5. Review Comments to the Author

Reviewer #1: The manuscript entitled “Study on the impact of scientific and technological talents on high-quality development of manufacturing industry: An empirical study based on provincial panel data” provides an empirical analysis on studying the relationship between scientific and technological talents and high-quality development of manufacturing industry, using the two-way fixed effects model and the spatial econometric model as the identification strategies. Also, the authors further explore two underlying mechanisms through which scientific and technological talents can affect high-quality development of manufacturing industry, i.e., consumption structure upgrading and productive service industry aggregation. Overall, the manuscript is interesting. I have several minor comments and suggestions for the authors:

1. The title of the article is too lengthy, resulting in poor comprehensibility. Thus, I would recommend the authors to modify the title of the manuscript as “A pathway towards high-quality development of manufacturing industry: Does scientific and technological talent matter?”

2. The abstract should provide a highly concise introduction to the research background, methods, findings, and significance of the article. However, I cannot see these items clearly in the current version. Also, the research gap is not identified in a compelling way in the introduction section.

3. English language and grammar issues are too abundant throughout the manuscript, and need a professional proofreading. For example, see the following sentence:

“Based on human capital theory, endogenous growth theory, and industrial agglomeration theory to construct a theoretical analysis framework, panel data from 30 provinces in China from 2012 to 2021 were used as research samples to measure the level of scientific and technological talents and the level of high-quality development of the manufacturing industry by the entropy weight method, and a two-way fixed effect model, a mediated effect model, and a spatial Durbin model were used to empirically investigate the influence of scientific and technical talents on the high-quality

development of the manufacturing industry”

This is only one example, and similar issue are abundant.

4. Comparisons with other studies have to be provided in the DISCUSSION section. Please interpret and describe the significance of your findings in light of what was already known about the research problem being investigated and explain any new understanding or fresh insights about the problem after you've taken the findings into consideration. Please provide a comparison with other studies.

5. Its topic appears to be within the scope of carbon finance or sustainable finance, the authors should cite research from PLOS ONE journal to connect to this ongoing discourse.

6. The last section needs to be re-organized, with more insightful implications of your findings for managers and researchers (ideally in two separate sections, i.e., theoretical and practical implications). Also, please elaborate on more limitations and future research directions.

Good luck!

6. PLOS authors have the option to publish the peer review history of their article (what does this mean?). If published, this will include your full peer review and any attached files.

Reviewer #1: No

---

## [Author Response · Author response to Decision Letter 0]

13 Aug 2023

Response to journal comments：

1.Based on the journal's template, We have made the first changes to the manuscript's typography, fonts, and file name.

2. Changes have been made to the Acknowledgments section of the manuscript by removing the reference to funding information. In addition, no changes are required to the Funding Statement of this paper, which we have explained in the Appendix.

3. Concerning the role of funders in the financial disclosure section, funders have a role in the study, and we have described the role of funders in the cover letter.

4. we do not need to change the data availability statement.

5. We have made corrections to the personal information as prompted by you by linking ORCID iD to my Editorial Manager account.

6. we have corrected the submission information as you requested by adding the author Li Dan.

7. We removed the icons from the manuscript and uploaded the icons in TIF format. 

8. The tables in the manuscript have all been mentioned in the text, and the revised parts have been marked in red in the manuscript. 

9. We have added the supporting information section at the end of the article and uploaded the relevant documents.

Responses to reviewers' comments：

1.The title of the article has been revised in accordance with the recommendations of the reviewers.

2.In the abstract section, the content was revised to present the background, methodology, results, and significance of the article. In the introduction section, the limitations of the existing research and the contributions of the article were presented in the last two paragraphs of the introduction, respectively, highlighting the gaps and innovations between the article and the existing research results.

3.We have professionally proofread the entire article to correct any grammatical problems.

4.In the discussion section, a comparison of the results of this study with other studies is included, and the significance of the findings and the understanding of the research questions are elaborated, as described in the concluding sections of articles 4.1, 5.1, 5.2 and 5.3.

5.We have included in the references articles from PLOS ONE journals in the relevant categories.

6.We have reorganized the last part of the article to include the significance of the findings and the limitations and future research directions of the article. See articles 6.2 and 6.4 for details.

---

## [Decision Letter · Decision Letter 1]

10 Nov 2023

A pathway towards high-quality development of  the  manufacturing industry: Does scientific and technological talent matter?

PONE-D-23-20007R1

Dear Dr. LI,

We’re pleased to inform you that your manuscript has been judged scientifically suitable for publication and will be formally accepted for publication once it meets all outstanding technical requirements.

Kind regards,

William Mbanyele, PhD

Academic Editor

PLOS ONE

Additional Editor Comments (optional):

Reviewers' comments:

Reviewer's Responses to Questions

**Comments to the Author**

1. If the authors have adequately addressed your comments raised in a previous round of review and you feel that this manuscript is now acceptable for publication, you may indicate that here to bypass the “Comments to the Author” section, enter your conflict of interest statement in the “Confidential to Editor” section, and submit your "Accept" recommendation.

Reviewer #1: All comments have been addressed

2. Is the manuscript technically sound, and do the data support the conclusions?

Reviewer #1: Yes

3. Has the statistical analysis been performed appropriately and rigorously? 

Reviewer #1: Yes

4. Have the authors made all data underlying the findings in their manuscript fully available?

Reviewer #1: Yes

5. Is the manuscript presented in an intelligible fashion and written in standard English?

Reviewer #1: Yes

6. Review Comments to the Author

Reviewer #1: The revised manuscript has been significantly improved and I do not have further comments

7. PLOS authors have the option to publish the peer review history of their article (what does this mean?). If published, this will include your full peer review and any attached files.

Reviewer #1: No

---

## [Editor Report · Acceptance letter]

2 Jan 2024

PONE-D-23-20007R1 

PLOS ONE

Dear Dr. LI, 

I'm pleased to inform you that your manuscript has been deemed suitable for publication in PLOS ONE. Congratulations! Your manuscript is now being handed over to our production team.

Kind regards, 

on behalf of

Dr. Ming Zhang 

Academic Editor

PLOS ONE